# EMDR–Teens–cPTSD: Efficacy of Eye Movement Desensitization and Reprocessing in Adolescents with Complex PTSD Secondary to Childhood Abuse: A Case Series

**DOI:** 10.3390/healthcare12191993

**Published:** 2024-10-06

**Authors:** Julie Rolling, Morgane Fath, Thomas Zanfonato, Amaury Durpoix, Amaury C. Mengin, Carmen M. Schröder

**Affiliations:** 1Department of Child and Adolescent Psychiatry, Strasbourg University Hospital, BP 426, 67091 Strasbourg, Franceschroderc@unistra.fr (C.M.S.); 2Regional Center for Psychotrauma Great East, Strasbourg University Hospital, BP 426, 67091 Strasbourg, France; amaury.mengin@chru-strasbourg.fr; 3National Center for Scientific Research (CNRS), Research Unit 3212, Sleep, Clock, Light and Neuropsychiatry, Institute for Cellular and Integrative Neurosciences, BP 426, 67091 Strasbourg, France; 4Faculty of Medicine, Strasbourg University, 67000 Strasbourg, France; thomas.zanfonato@chru-strasbourg.fr (T.Z.); amaury.durpoix@chru-strasbourg.fr (A.D.); 5Department of Psychiatry, Mental Health and Addictology, Strasbourg University Hospital, BP 426, 67091 Strasbourg, France; 6National Institute of Health and Medical Research (INSERM) Unit 1329, Strasbourg Translational Neurosciences and Psychiatry, BP 426, 67091 Strasbourg, France

**Keywords:** adolescent, physical and sexual abuse, EMDR, post-traumatic stress disorder (PTSD), complex post-traumatic stress disorder (cPTSD)

## Abstract

**Background**: Mental healthcare for children and adolescents with a history of childhood abuse constitutes a major public health issue. Indeed, abuse exposes children to severe and complex post-traumatic stress disorder (cPTSD) but also to neurodevelopmental and psychological repercussions impacting the developmental trajectory. Trauma-focused care is essential to avoid the chronicization of symptoms and disorders. **Objective**: The aim of this prospective case series study was to investigate the efficacy of eye movement desensitization and reprocessing (EMDR) on complex post-traumatic symptoms and associated psychiatric disorders in adolescents with a history of abuse. **Method**: Twenty-two adolescents, aged 12 to 17, who had been abused during childhood were included. All adolescents met ICD-11 criteria for complex PTSD. Subjective measures of PTSD and associated psychiatric disorders were taken before (T0) and after 3 months of EMDR therapy (T1). **Results**: The average PTSD symptom score on the CPTS-RI significantly decreased from 40.2 to 34.4 after EMDR, indicating improvement in post-traumatic symptoms. A significant decrease in the average depression score (CDI from 18.2 at T0 to 10.6 at T1), anxiety score (R–CMAS from 21.3 at T0 to 13.3 at T1), emotional regulation score (ALS from 29 at T0 to 10.8 at T1), insomnia score (ISI from 18.5 at T0 to T1 of 9.2 at T1), and harmful use of alcohol and drugs score (ADOSPA from 2.3 at T0 to 0.3 at T1) was observed after EMDR therapy, as well as an increase in quality of life (CBCL 4–16 score from 57.9 at T0 to 77.4 at T1). **Conclusions**: The results of this study are encouraging and suggest that EMDR may be effective in the symptom management reducing post-traumatic symptoms and certain comorbid disorders frequently seen in adolescents who have experienced childhood abuse. Further research is needed on adolescent populations suffering from cPTSD (e.g., randomized controlled trials with control groups and other therapies or evaluating the action of the different phases of the study).

## 1. Introduction

### 1.1. Child Abuse

In Europe and around the world, the abuse of children during childhood is a phenomenon that is far from being controlled. To date, the World Health Organization (WHO) estimates the prevalence of maltreatment in Europe at 29.1% for psychological violence, 23% for physical violence, and 9.6% for sexual violence (World Health Organization, 2022) [1]. Epidemiological research on maltreatment and adverse life events complements the historical work of Felitti and shows that in the United States, 66% of a sample of 4053 young people (aged 2 to 17 years) report having been exposed to more than one form of victimization during their lifetime [2], and 22% of a nationally representative sample of 2030 children (aged 2 to 17 years) report having suffered at least four different forms of victimization in the past year [3]. Concerning the form of maltreatment, 15.2% of a population of 4000 children (aged 0 to 17 years) declared having been victims of maltreatment by the person who took care of them with 5.8% who would have witnessed domestic violence between parents, 5% reporting having suffered physical violence, and 2% of girls reporting having suffered sexual abuse (National Survey of Children’s Exposure to Violence) [4]. In France, when asked about their own childhood, 22% of a French sample (Harris Survey, 2017; sample of 1030 French people) reported events comparable to abuse with 16% of these self-declared victims testifying to abuse of a sexual nature (mainly touching young girls), 8% reporting psychological abuse (threats, insults, humiliation), 5% reported regular violence (beatings), and 3% reported repeated neglect. 

### 1.2. Abuse and Complex PTSD

The effects of often log-term abuse among young people are divers. Scientific publications highlight consequences in different areas of functioning, including dependence and attachment disorders [5], emotional difficulties, internalizing and externalizing problems such as aggressiveness and antisocial tendencies [5,6], and poor school adjustment [5,6]. The functional consequences resulting specifically from trauma are often considered to meet the diagnosis of post-traumatic stress disorder (PTSD) or, more recently, complex post-traumatic stress disorder (cPTSD). cPTSD is characterized by PTSD associated with severe and persistent disturbances *in* the organization of the self (POS) with (i) emotional regulation difficulties, (ii) alterations in self-perception, and (iii) relational difficulties [7]. This diffuse and multifaceted symptomatology is classically defined as secondary to repeated interpersonal traumatic exposure, from which it is difficult to escape, particularly following childhood abuse. PTSD and cPTSD are not the only possible psychological consequences of abuse, and the functional repercussions of abuse can also result in other mental disorders (depression, anxiety disorders) or behavioral manifestations (behavioral disorders, social behavior disorders, suicide attempts, substance use) [8,9].

To date, the emerging literature on cPTSD estimates the prevalence of cPTSD between 2.4% (for children aged between 8 and 17 exposed to a single traumatic event) [10] and 3.4% (for young people aged 11 to 19 exposed to at least one traumatic event, compared to 1.5% of PTSD in the same population) [11]. Overall, young people with cPTSD who received social care through the child protection system following maltreatment present more mental health disorders, more risk of developing more severe and chronic post-traumatic symptoms as well as higher levels of risky behaviors or social dysfunction [12] than children who suffered single traumas causing PTSD (a rate of 15.9% PTSD was observed in a child psychiatry population 1 year after exposure to a single event [13]). 

Ultimately, young people with cPTSD die prematurely with a 25 years shorter life expectancy, which explains why the care of children and adolescents with cPTSD is widely recognized as a major public health issue [1]. For these young people, early care targeting psychotrauma aims to avoid the perpetuation of the disorders [14] and also the negative effects on neurodevelopment [15] and the construction of personality [8,9], as without therapy targeted at psychotrauma, symptoms can persist for several years [16]. Effective and well-tolerated therapies [14,17] do exist for this age group but still need to be validated, specifically in cPTSD in children and adolescents.

### 1.3. Validated Therapies for the Treatment of Psychotrauma

Therapeutic approaches for cPTSD initiate an intense and complex process aiming at improving life trajectories punctuated by breakups, self-destructive behaviors (e.g., suicide attempts, risky behavior, endangerment, addictions), and revictimization. The therapeutic needs of these adolescents are significant and evolving, involving multimodal care with a range of varied interventions targeting all areas of functioning [14,18] and a flexible intensification of care when these young people face challenges throughout adolescence or when they are subjected to additional stressors.

Internationally, Australian [19] and NICE [20] guidelines recommend EMDR therapy in the treatment of PTSD, while the APA suggests the use of EMDR [21]. Thus, even if there is a strong trend in favor of the use of EMDR, these recommendations differ and mainly concern adults and PTSD treatment. Indeed, cPTSD is a relatively new diagnosis, for which there are currently few specific recommendations regarding its management. As such, the recommendations for the treatment of cPTSD of the International Society for Traumatic Stress Studies (ISTSS) [14] recommend a sequential and multimodal approach with a first phase of stabilization of the symptoms of self-organization disorders called Skills Training in Affective and Interpersonal Regulation for Adolescents (STAIR-A) [22] and a second phase specifically dealing with trauma. For the second phase, the ISTSS advocates the adaptation to complex trauma of therapies specifically validated for PTSD (such as CBT and EMDR), despite a limited effect on organization of the self-symptoms. These recommendations are motivated by the fact that these therapies contain elements treating both PTSD symptoms and comorbid disorders (depression, anxiety, and substance use disorders) frequently associated with cPTSD [14]. To date, the ISTSS mainly emphasizes trauma-focused CBT (Tf-CBT) in child psychiatry populations and less in EMDR [14], while the Australian guidelines cautiously advance the presumed usefulness of psychological interventions focused on trauma. As it currently stands, the imprecision of the recommendations for children and adolescents confirms the need to increase knowledge of the efficacy of EMDR in young people with complex trauma [19]. 

### 1.4. EMDR in Children and Adolescents

EMDR combines different well-established psychotherapeutic techniques such as imagined exposure, self-regulation, resource development, and cognitive change. It is a psychotherapeutic method based on the ‘adaptive information processing’ model (AIP). This model postulates that information related to the traumatic event would have remained stored in a dysfunctional manner in the traumatic memory, explaining their intrusive and emotionally distressing nature [23]. The originality of this therapy is the use of alternating bilateral stimulation (SBA) to reprocess dysfunctional post-traumatic contents. 

The standard EMDR therapy protocol includes eight phases with (1) a phase of collecting the patient’s history (establishment of the therapeutic alliance and verification of the indication for EMDR therapy), (2) a preparation phase (learning emotional stabilization and self-regulation and body anchoring techniques), (3) the selection of the image to be processed (target image) and the search for negative cognition (CN) associated with this image as well as the positive cognition (CP) that we wish to obtain, (4) a desensitization phase with the realization of alternating bilateral stimulations (SBA) which allows the trauma to be reprocessed, (5) the integration of positive beliefs to strengthen self-esteem, (6) a body scan, and finally, the (7) closure and (8) re-evaluation of the effectiveness of the reprocessing [23]. According to the practitioners, EMDR is currently either considered the main treatment for cPTSD or as a complementary treatment included in a sequential and multimodal care of the cPTSD treatment. In our study, we consider phases 1 and 2 as preparation phases for which we have reinforced stabilization exercises (four sessions), and phases 3 to 8 as EMDR treatment phases (six sessions) (see intervention procedure).

Over the past twenty years, several systematic reviews and meta-analyses have highlighted the effectiveness of EMDR for the treatment of PTSD in children and adolescents [24,25,26]. Nevertheless, while studies evaluating the effectiveness of EMDR on pediatric populations with single traumatic exposure [27] provide strong evidence of its effectiveness on PTSD and its comorbidities [28], few studies have specifically investigated EMDR treatment in children with cPTSD, as recommended by national and international guidelines [14]. To date, only three controlled studies have evaluated the effectiveness of EMDR in children who have suffered multiple traumas [29]. Of these three studies, one focused mainly on post-traumatic symptomatology [30] while the other two were more specifically interested in conduct disorders linked to complex trauma [31,32]. Given (i) the demonstration of the effectiveness of EMDR on PTSD and its comorbidities, (ii) the consequences of abuse on the child’s development, (iii) the chronicization risk and especially the fact (iv) that post-traumatic reactivation and disruption in interpersonal relationships frequently lead these young people to avoid care, EMDR should be further studied specifically for cPTSD in children and adolescents with a history of physical or sexual abuse.

### 1.5. Study Objectives

Our prospective case series study, EMDR–Teens–PTSD, was designed to evaluate the effectiveness of a series of four preparation sessions and six sessions of EMDR on PTSD symptomatology. The secondary objectives of the study were to evaluate the effectiveness of EMDR on the main comorbidities associated with adolescent PTSD symptoms such as depression, anxiety, emotional regulation disorders, sleep disorders, behavioral disorders and risky behavior, and illicit substance consumption but also its effectiveness on quality of life and parental anxiety. The tertiary aims of the study were to develop proposals for adjustments to future EMDR protocols for adolescents suffering from post-traumatic stress disorder, with particular reference to work with parents.

## 2. Materials and Methods

### 2.1. Study Design and Participants 

To be recruited into the EMDR–Teens–cPTSD study, adolescents had to meet the diagnostic criteria of cPTSD according to ICD-11 [7]. For ICD-11 cPTSD criteria, adolescents had to present all of the symptoms of PTSD ((i) *re-experiencing the traumatic event* or events in the present in the form of vivid intrusive memories, flashbacks, or nightmares; (ii) *avoidance* of thoughts and memories of the event or events or avoidance of activities, situations, or people reminiscent of the events; and (iii) *persistent perceptions of heightened current threat*) associated with severe and *persistent disorders of self-organization (POS),* including symptoms of (i) *deregulation severe* and pervasive affects, (ii) *negative self-perception*, and (iii) permanent *difficulties in interpersonal relationships* [7]. These criteria had to be present after an extremely threatening or horrific event or series of events and the symptoms had to persist for at least several weeks and cause significant impairment in personal, family, social, educational, occupational, or other important areas of functioning. A double clinical evaluation was carried out by clinicians specializing in psycho-traumatology from the Regional Psychotrauma Center in the Grand-Est region of France.

From September 2019 to July 2022 (excluding the period from March 2020 to October 2021 due to the COVID-19 pandemic), 22 adolescents (aged 12 to 17 years) with cPTSD following childhood abuse were included. In addition to diagnostic criteria (see above), adolescents had to (i) be able to express support for participation in the research and (ii) have obtained parental consent. Furthermore, the adolescents included had to benefit from (iii) child psychiatric monitoring. The exclusion criteria were (i) having severe developmental delays, (ii) having psychotic disorders, (iii) participating in another research study evaluating the effectiveness of psychotherapy targeting PTSD or cPTSD, or (iv) participating in any other study evaluating the effectiveness of a drug treatment for PTSD or cPTSD.

EMDR psychotherapy included six individual sessions carried out by a therapist pair from the Child and Adolescent Psychiatry Department (CAPD) of Strasbourg University Hospitals. EMDR therapy was offered to young people who had previously benefited from child and adolescent psychiatric care (day hospital and psychological monitoring) for at least 3 months.

### 2.2. Measures

#### 2.2.1. Socio-Demographic and Clinical Characteristics of Adolescents and Their Families

All sociodemographic data of adolescent subjects (age, sex, place of accommodation) were extracted from a medical–administrative document completed upon admission to the Child and Adolescent Psychiatry Department. Medical clinical interviews assessed traumatic exposures such as nature of traumatic events, date and frequency of traumatic exposure, and parental history of exposure to traumatic events. Information on family situation and protection measures (history of protection measures or current protection measures) were also collected.

#### 2.2.2. Assessment of Adolescent PTSD Symptoms 

The effectiveness of EMDR on post-traumatic symptoms and comorbid disorders was evaluated using validated self-questionnaires completed by the adolescent and by one parent when available. The initial assessment was carried out one week before therapy (T0), and the post-treatment assessment (T1) was carried out 3 months after the end of the EMDR therapy (after the 6th EMDR session).

The impact of EMDR on post-traumatic symptoms was assessed at T0 and T1 using (i) the Child Post-Traumatic Stress Reaction Index (CPTS-RI) and the Peri-Traumatic Dissociative Experiences Questionnaire (PDEQ). The Child Post-Traumatic Stress Reaction Index (CPTS-RI) is a 20-item self-questionnaire used to confirm the diagnosis of PTSD and its level of severity. A total score of 12–24 is associated with mild PTSD, a score of 25–39 with moderate PTSD, a score of 40–59 with severe PTSD, and a score above 60 with very severe PTSD. The Cronbach alpha coefficient is 0.87 for the French version of the CPTS-RI [33]. The Peritraumatic Dissociative Experiences Questionnaire (PDEQ) [34] measures the experiences of dissociation experienced with 10 items assessing the degree of depersonalization, unreality, amnesia, the modification of time perception, and the modification of one’s body image. A score greater than 15 indicates significant peri-traumatic dissociation.

#### 2.2.3. Assessment of PTSD Comorbidities in Adolescents and Parents

The impact of EMDR (difference at T0 and T1) on PTSD comorbidities was assessed using (ii) the Child Depression Inventory (CDI) to quantify depressive symptoms (scores ranged from 0 to 54; the higher the score, the more severe the level of depression [35]), (iii) the Revised Children’s Manifest Anxiety Scale (*R*-*CMAS*) to quantify anxiety symptoms [36], (iv) the Affective Lability Scale (ALS) to quantify emotional regulation difficulties [37], (v) the Insomnia Severity Index (ISI) to assess insomnia (total score specifies the degree of insomnia: absence of insomnia (0–7), symptoms of subclinical insomnia (8–14), moderate insomnia (15–21) and severe insomnia (22–28)) [38], (vi) the Child Health Questionnaire CF87 (CHQ-CF87) to assess different dimensions (physical functioning, emotional state, physical limitations, pain, behavioral limitations, overall behavior, mental health, self-esteem, perception of health, change in state health, family activity and family cohesion) of quality of life (dimensions scored on a scale ranging from 0 (worst possible quality of life) to 100 (best possible quality of life)) [39], (vii) the Adolescents and Psychoactive Substances (ADOSPA) scale to quantify the harmful use of alcohol and drugs (low risk, 0–1 points; moderate risk, 2 points; and high risk, ≥3) [40] and (viii) the Child Behavior Checklist 4–16 (CBCL 4–16) to evaluate the child’s behavior based on parental observation (sensitivity threshold score at 40 for boys and 37 for girls) [41]. Parental anxiety was assessed with (xi) the administration of the State–Trait Anxiety Inventory (STAI) [42] in order to obtain indirect information on the parents’ psychological state and their possible possibilities for emotional regulation. The total score indicates the severity of the general state of anxiety with a rating of 20 to 80. A score between 36 to 45 indicates low anxiety, a score between 46 and 55 indicates moderate anxiety, a score between 56 to 65 indicates high anxiety and a score above 65 indicates very high anxiety.

### 2.3. Intervention Procedure

All 22 adolescent subjects included in the study benefited from a total of four preparation seance and six EMDR sessions at a rate of one session of 90 min per week (total treatment: TT “4 + 6”. The therapy was carried out by an EMDR-certified psychologist and child psychiatrist. Therapists also benefited from bimonthly supervision sessions. In our study, we used the developmental EMDR protocol [43] suitable for adolescents over 12 years old. This protocol is similar to the adult protocols while integrating elements of the children’s protocol (intended for those under 12 years old), with its content and session duration as well as the integration of caregivers being adapted to the level of development, complexity of the trauma and caregiver quality (family dynamics, type of attachment, parental availability) (see arrangements proposed by B. Tinker) [43,44].

#### Adaptations of the EMDR Protocol for Adolescents with cPTSD

The complex nature of psychotrauma presented by our adolescent population also justified the establishment of a therapeutic protocol reinforcing the preparation phases (4 sessions for learning emotional stabilization and self-regulation and body anchoring techniques) before the 6 traumatic treatment sessions. During these preparation stages, our objective was to establish a reassuring and secure therapeutic bond to engage in care by helping the adolescents to identify and strengthen their resources and to improve their emotional tolerance by targeting dissociative tendencies. With this in mind, we worked on (1) regaining control through acquisition of knowledge about oneself (psychoeducation); (2) security through exercises on a ‘safe place’, on distancing and on the feeling of self-control (such as the imagery of a protective screen); (3) the development of awareness of present time with bodily anchoring; and (4) the parents’ ability to support the adolescent’s distress and to be a ‘safe place’ for the child. In terms of traumatic reprocessing through alternating bilateral stimulations (in our study, mainly ocular and auditory), we used (i) the oscillation technique, allowing us to stay within the tolerance window by alternating exposure and return to a zone of calm (double attention to trauma and current solutions), and (ii) the reverse protocol (future–present–past), allowing the creation of resources by projecting positively into the future [45]. In the event of significant activation, we utilized the ‘STOP’ signal, and we used several types of stimulations simultaneously in order to maximize the load on working memory. Cognitive distortions were worked on using approaches focusing on repair (repair scenarios) or reparenting techniques aimed at developing patterns of self-care.

### 2.4. Procedure, Legal and Ethical Framework

The study was carried out in accordance with the research regulations set out in the Declaration of Helsinki. For this study, an agreement was established between the University Hospitals of Strasbourg and the child protection home placement services. The study was approved by the ethics committee of the University of Strasbourg (CE-2024-54).

All data were collected using validated standardized assessment instruments, according to ongoing routine monitoring procedures at the Department of Child and Adolescent Psychiatry. Participant satisfaction was assessed using visual analog scales (continuous variables ranging from 0—dissatisfaction to 10—very satisfied). Adherence to treatment (EMDR completion rate) and feedback from professionals were also integrated.

### 2.5. Data Analysis

Statistical analyses were carried out using Jamovi software *(Version 2.5.3)*. The demographic and clinical characteristics of participants were examined using descriptive statistics. We used the mean ± standard deviation (M ± SD) for numerical variables and the number and percentage (%) for categorical variables. The small number of subjects (*n* = 22) and the absence of Gaussian distribution justified the use of the Mann–Whitney test as the non-parametric method to compare pre- and post-therapy effectiveness. The significance threshold of 5% was retained, corresponding to *p* < 0.05.

## 3. Results

Twenty-two adolescents underwent six weekly EMDR sessions for 90 min per session (see their socio-demographic characteristics in Table 1).

### 3.1. Initial Psychiatric Assessment

Before treatment (T0), our population of 22 adolescents displayed a mean score of 40.2 (±4.6) on the CPTS-RI (see Table 2). Subtype analysis shows that 55% of the population presented severe PTSD (CPTS-RI: 40–59) and 45% of the population presented moderate PTSD (CPTS-RI: 25–39). Our sample presented a peri-traumatic dissociation score of 25.7 (±3.3) on the PDEQ (significance threshold greater than 15), which corresponds to moderate peri-traumatic dissociation.

### 3.2. Evaluation of the Effectiveness of EMDR Treatment

Acceptability of EMDR treatment as measured by visual analog scales was on average 96%. Only one patient did not finish treatment due to intercurrent events impeding the continuity of care (serious medical pathologies of both parents) associated with a minimization of PTSD by the adolescent’s family.

The mean PTSD symptom scale score significantly decreased from 40.2 (±4.6) to 34.4 (±5.4) after EMDR (see Table 2), indicating improvement in post-traumatic symptoms ranging from severe to moderate in intensity. No patient experienced a worsening of PTSD symptoms. A significant reduction in scores for depression (mean score of 18.2 (±3.6) at T0 and 10.6 (±5.4) at T1, *p* < 0.05), anxiety (mean score of 21.3 (±5.5) at T0 and 13.3 (±6.3) at T1, *p* < 0.001), insomnia (mean score of 18.5 (±4.4) at T0 and 9.2 (±5.4) at T1, *p* < 0.001) and harmful use of alcohol and drugs (mean score of 2.3 (±1) at T0 and 0.3 (±0.7) at T1, *p* < 0.001) was observed after EMDR therapy. Furthermore, an improvement in emotional regulation (ALS mean score of 29 (±11.3) at T0 and 10.8 (±9) at T1, *p* < 0.05) and quality of life (see below) was also observed. On average, emotional lability decreased by 18.2 points on the CDI with high variability (standard deviation of 11.3 at T0 and 10.8 at T1). All measured dimensions (depression, mood elevation, anxiety, anger, anxiety, depression and bipolarity) improved after EMDR.

Improvement in the quality of life score (CHQ-CF 87) (mean score from 57.9 (±9.8) at T0 to 77.4 (±13.6) at T1) was seen in all areas of operation and was particularly pronounced for factors such as emotional state (mean score of 68.7 (±14.1) at T0 to 93.3 (±4.8) at T1), family cohesion (mean score of 51.6 (±12.1) at T0 to 90.3 (±5.4) at T1) and family activities (mean score from 38.3 (±4.7) at T0 to 64 (±11.6) at T1 (see Figure 1).

Behavioral problems on the CBCL 4–16 decreased by 22.4 points (non-significant due to high variability (*sd* = 25.9 at T0; *sd* = 24.8 at T1)). Indeed, all categories of behavioral disorders are improved after EMDR therapy apart from the somatic complaint’s subcategory (mean score of 4 at T0 and 4.8 at T1).

Of the 72% of parents who completed the questionnaire, 66.7% had moderate trait anxiety (greater than 46) and 33.3% had low trait anxiety. Finally, there was no difference in intra-group efficacy between adolescents treated before and after COVID-19.

## 4. Discussion

Our EMDR–Teens–cPTSD study is one of the first studies to evaluate the impact of EMDR therapy on the overall post-traumatic symptomatology of 22 adolescents with complex PTSD following maltreatment [29,30,31,32]. Study results suggest that EMDR is an effective method for reducing (i) post-traumatic symptoms of adolescents with complex trauma; (ii) other comorbid disorders such as anxiety, depression and the harmful use of alcohol and drugs; (iii) overarching non-specific symptoms of complex PTSD (such as insomnia or emotional dysregulation); and that this therapy improves (iv) the quality of life of adolescents with cPTSD. 

### 4.1. Comparison with Data from the PTSD Literature

The adolescents included in this study were mainly girls with severe PTSD (see Table 1 and Table 2) associated with a significant degree of dissociation and numerous comorbidities. Female overrepresentation (73%) is common in the literature. Some authors explain this phenomenon by overexposure to certain traumatic events such as sexual violence, as seen in our study, while other authors suggest lower resilience capabilities in girls [46,47,48]. Beyond gender, early exposure to abuse (82% before the age of 5), the repetition of trauma over time, and the fact that the aggression was mainly committed by an attachment figure (86%) are all elements that have been associated with worse prognosis, being risk factors for complex PTSD associated with different comorbidities [49]. The link between dissociation and psychotrauma has been the subject of numerous publications showing that dissociation is associated with the risk of developing PTSD and higher severity of PTSD, and that in situations of early, chronic and severe trauma, as is the case in our study, dissociation is more severe [50]. Furthermore, associated attachment disorders have also a serious impact on the prognosis since attachment disorders (i) are highly common in maltreated children [8,51] and (ii) constitute one of the main risk factors for developing PTSD or other psychiatric disorders following childhood violence [52,53]. Additionally, attachment disorders (insecure and disorganized attachment) increase vulnerability to dissociation both in traumatic situations but are also independent of any traumatic exposure [54,55]. Thus, even if we did not specifically explore attachment disorders in our study population, 55% of the adolescents in our study were in foster or institutional care. Although it is difficult to conclude whether the severity of PTSD and associated disorders in our participants is attributable to traumatic exposures or attachment difficulties, it is likely that both components impacted the development of PTSD and its severity. 

Concerning the impact of EMDR on post-traumatic symptomatology, our results are in line with those of the latest studies carried out in adolescents with PTSD [56] and studies carried out with adolescents who have been exposed to repeated trauma resulting in complex trauma [29] or conduct disorders [31,32]. Thus, our results are comparable to the work of Jaberghaderi, who shows a significant reduction in post-traumatic symptomatology after EMDR treatment in 14 young Iranian girls (aged 12–13 years) who were victims of sexual abuse (a CROPS pre-test score of 34.86 versus a 18.86 post-test score) [30]. In this study, the authors suggest an average number of 6.1 sessions for EMDR to be effective, compared to 11.6 sessions for CBT, which may explain the similarity of the results (six sessions in our protocol). Other studies demonstrated an even greater reduction in post-traumatic symptoms with the average CPTS-RI score decreasing from 60 (±8.7) to 24 (±10.1) [56]. These differences may be explained by the time of the evaluation (3 months after traumatic exposure in our study compared to 6 weeks in the study of Karadag) and by differences in the study population (63% unique traumatic exposures for Karadag et al., compared to 14% only in our study). In this respect, the population of Farkas et al.’s work is similar to our population, since it focused on 40 adolescents (including 73.7% girls) cared for by youth protection services who suffered multiple traumas. In this article, Farkas et al. demonstrated a significant improvement in post-traumatic symptoms and behavioral problems with the effect maintained 3 months after the end of EMDR therapy, as well as a significantly greater reduction in depression and anxiety scores. However, EMDR was combined with a motivation–adaptive skills–trauma resolution (MASTR) approach, so it is difficult to disentangle improvements specifically related to the EMDR component of the study versus the MASTR. Another study carried out in 20 boys ages 10–16 suffering from conduct disorders following multiple traumatic exposures reported a significant reduction in PROPS scores (post-traumatic symptoms reported by parents) after treatment and a tendency towards a reduction in IES (Impact of the Event Score) scores at 2 months in participants who received EMDR compared to the treatment-as-usual group [31]. These trends could be explained by the reduced number of EMDR sessions which amounted to three in this study compared to six in our study. 

Our results provide additional arguments in favor of an effective duration and acceptability of EMDR therapy over a period of 3 months [32]. Indeed, the low dropout rate of our participants (4% in our study) is comparable to other studies [29,57,58]. This rate indicates good acceptability, suggesting that the adjustments to the protocol (four preparation sessions followed by six EMDR reprocessing sessions) are suitable for adolescents with cPTSD, although there is currently no consensus on the optimal number of EMDR sessions (interventions varying from two to twelve sessions depending on the protocols) for cPTSD treatment [29]. As such, some authors suggest that at least 12 or 20 sessions are necessary to obtain lasting improvement in POS symptoms beyond the simple reduction of PTSD symptoms [59], while other teams advocate approaches such as stabilization (integrated or not with trauma treatment) to deal with these disturbances [14,29]. In the future, it will therefore be essential to evaluate the differential impact of EMDR and stabilization phases (associated or not) on PTSD symptoms, POS and on attachment. Finally, even if desensitization and processing of traumatic memories may activate the patient [60], the absence of detailed verbalization of the traumatic [61] and the speed of the therapeutic response [18,62,63] could explain the excellent acceptability of the therapy in patients known for their difficulties in initiating and continuing care. Indeed, the risk of disruption of care by young people with cPTCD necessitates the development of brief [64], intensive [65] and adaptable programs, allowing the patient to choose the traumatic content to work on, namely physical perceptions, emotions or representations.

### 4.2. Comparison with Data from the PTSD Comorbidities Literature

Concerning cPTSD comorbidities, our results on the impact of EMDR on depressive symptomatology are in line with previous studies on young people repeatedly exposed to trauma [32] and young people with PTSD but without apparent symptoms of disturbance of self-organization [27,66], although more research is needed for these populations since the latest meta-analysis by Morena-Alcazar indicates only a trend toward reduction in depressive symptoms [26]. Furthermore, the reduction in the anxiety of our participants is consistent with research carried out in young people presenting with cPTSD [67] symptoms following repeated [32] and single traumatic exposures [26,27,56,66]. In our study, the reduction in anxiety is relatively modest (8 points on the R-CAMS), which can be explained by the fact that the parents in our population also presented trait anxiety and had few personal resources (82% of parents look after their child alone), which might have influenced the measured impact of EMDR therapy on anxiety symptoms. 

In our participants, EMDR had an positive and consistent impact on different interconnected areas of functioning such as emotional [68] and behavioral regulation [30,31,69], sleep, substance use and quality of life. As such, we suggest that EMDR constitutes a pivotal therapeutic tool allowing the integration of traumatic memories and improving emotional regulation, sleep and overall functioning. Furthermore, the beneficial effect of EMDR on quality of life was particularly notable in categories related to the family: family activity, family cohesion and emotional state (see Figure 1). These improvements may be linked to improved emotional and behavioral regulation and their impact on the parent–child relationship, since in this study adolescents reported EMDR therapy with parental participation (76% of parents), which made it possible to (i) reduce reactivations, (ii) strengthen security, (iii) increase the mobilization of the adolescent’s personal resources and (iv) promote parent–child harmony [70]. At the same time and despite initial reluctance (parental history of violence, painful experiences with child protection), parents have improved (i) their belief in their child’s abilities, (ii) the validation of the child’s experience, (iii) tolerance of their child’s emotions and (iv) management of their own emotional reactions [71].

### 4.3. Therapeutic Work with Parents 

Parental involvement and support is known to constitute a strong protective factor for children [72] and a key to the success of psychotherapeutic treatments overall. In our study, we chose to integrate parents into care through parental guidance [27] with an initial time dedicated to the evaluation of the child–parent bond and the quality of attachment, of family functioning and affiliations, as well as of parental investment (preparatory phase). During this stage, we were vigilant to ensure that the therapeutic objectives centered on the child by formalizing the presence of parent as a safe person serving as emotional support and a ‘guardian of integration’ of the child’s experience. Elements of psychoeducation and parenting skills training (such as working around cognitive distortions around the reprocessing of the child’s trauma) allowed parents to help their child stay within the window of emotional tolerance. At this stage, explanations of EMDR, advice regarding routine implementation and emotional regulation tools were widely accepted and applied by the family. EMDR treatment for trauma was carried out with the parents of the youngest adolescents (under 13), while the re-evaluation phase systematically involved the parents in order to validate the changes initiated and to work on the child–parent relationship. During this guidance work, we were able to observe real parental reinvestment, a destigmatization of family history and an openness to address transgenerational trauma.

### 4.4. Strengths of the Study

Our study is one of the first to evaluate the impact of EMDR therapy in adolescents presenting complex and severe post-traumatic symptomatology following maltreatment of different types. Our population is therefore representative of a clinical population of adolescent victims of child maltreatment, both in terms of traumatic exposure (frequent association of different types of traumatic exposure: physical, sexual and psychological maltreatment) and in terms of functional repercussions. Another strong point is the use of protocolized interventions carried out by certified, experienced and supervised clinicians to ensure the homogeneity of therapy delivery. Finally, the good acceptability (only one cessation of therapy) and the 100% completion rate of the questionnaires reinforces the solidity of our results. 

### 4.5. Study Limitations

This study is not without limitations. The absence of a control group in a study evaluating the efficacy of EMDR in adolescents with complex PTSD is the main limitation of our study. Indeed, without a no-therapy group, it is difficult to know whether the improvements observed are really due to EMDR’s effect or to the natural evolution of symptoms (spontaneous improvement bias), external factors (family support or others) or placebo effect, which limits the generalizability of the results and their scientific acceptance. However, the aim of the study was not to demonstrate superiority between therapies but rather to evaluate the efficacy of EMDR independently and on criteria specific to complex PTSD. Intra-group follow-up (before/after EMDR) with standardized measurement tools and longitudinal evaluation may be sufficient to demonstrate individual progress, intervention efficacy, acceptability and tolerance without the need for a control group. Furthermore, from an ethical point of view, it is questionable to deprive severely affected adolescents of validated treatment or to randomize them to a therapy they have not chosen. Forcing participation in a group delivering a therapy other than that preferred by the patient could result in less adherence to treatment. Finally, although Tf-CBT is a validated approach for the treatment of PTSD, it is not necessarily considered superior to EMDR in all studies, and specifically concerning the treatment of cPTSD (such as the Integrative Treatment of Complex Trauma, ITCT [73], no complex trauma therapy was available in the child psychiatric population in France at the time of the study. All in all, by focusing solely on EMDR, the study enables a direct assessment of its efficacy on a population and pathology of complex trauma with unique characteristics, while minimizing biases linked to ethical issues and non-adherence to treatment. The second limitation is the absence of long-term follow-up measures, which does not allow us to reach a conclusion on the maintenance of the beneficial effects of EMDR over time [74]. However, to compensate for this, we chose to evaluate the effectiveness of EMDR six weeks after the sessions in order to measure the durability of the therapeutic effects. The benefits of EMDR, notably the reprocessing of traumatic memories, can continue to be reinforced over time, beyond the end of the sessions. By avoiding the immediate effect of therapy, this delayed evaluation better captured therapeutic consolidation and the integration of learning into patients’ daily lives. It also identified whether the results persisted without the lingering effect of proximity to the therapist, ensuring that improvements are stable and autonomous, rather than influenced by immediate therapeutic attention. The third limitation is the use of subjective evaluations based on the adolescents’ feelings. Indeed, psycho-traumatized adolescents may have difficulty perceiving changes during therapy, and the use of self-questionnaires may lead to an underestimation of the disorders by the adolescent [75]. The fact that the questionnaire was administered in the presence of the therapist could also have created a bias in motivation and/or social desirability. Concerning the child psychiatry population under study, a combined analysis of subjective, objective (physiological reactivity markers and sleep markers) and observational effectiveness could have further improved our data. In this respect, to compensate for the current absence of a validated French version of a cPTSD assessment scale [76], we included adolescents meeting ICD-11 cPTSD criteria before assessing separately complex post-traumatic symptoms using the PTSD scale, its comorbidities and the main areas of functioning altered in cPTSD. (Please note that we are participating in the ongoing validation of the Children and Adolescent Trauma Screen 2 (CATS.2) coordinated by our colleagues in Lyon, Professor Fourneret and Dr Espi. ITQ validation project is underway by our team.) Finally, the confounding biases linked to the absence of environmental control (traumatic re-exposures, legal proceedings, family environment, etc.) [77] modulating the severity of PTSD symptoms and the absence of the assessment of attachment disorders frequently associated with cPTSD [53] make it difficult to attribute clinical improvement to the therapeutic EMDR intervention and the preparatory stabilization versus the work with parents. To better control some of these biases, we did not include participants presenting traumatic reactivation events (recent traumatic re-exposure, anniversary date of the trauma, trial) or participants for whom a major change in family organization was occurring (death, separation) [78]. Inclusion before and after COVID-19 could have been a limitation. However, the lack of differences between the results of the included and treated groups before and after COVID-19 was explained by the fact that all the adolescents already benefited either from ongoing access to care with therapeutic support or from a pre-existing socio-educational support network and that COVID-19 was experienced as a minimal stressor in relation to previous life events. This study is a preliminary study before a larger randomized controlled trial with a control group and blinding to group status.

## 5. Conclusions and Research Perspectives 

Young people with a history of childhood abuse suffering from complex PTSD constitute a particularly vulnerable population, frequently combining psychological disorders (post-traumatic symptoms, other psychiatric disorders and/or attachment disorders) and psychosocial difficulties (insecurity, parental psychopathology and dysfunctional family) that are chronic and invasive. Early and multimodal care is essential to prevent their developmental and psycho-affective trajectory from being permanently modified.

Our study is one of the first to show the rapid effectiveness of EMDR on PTSD symptoms and on the main associated comorbidities in adolescents with complex psychotrauma following abuse. This study provides interesting results for these populations whose treatment is complex since it must specifically treat traumatic memories, while reassuring and stabilizing adolescents presenting relational deficits and major cognitive distortions.

The results of these studies should be replicated in larger equivalent populations by evaluating the effects of EMDR to control groups (without or with therapies targeting cPTSD such as Integrative Treatment of Complex Trauma for Adolescents (ITCT-A)) and in the long term [73]. More generally, future research on EMDR in complex trauma should incorporate protocol adaptations focusing on standardizing (i) the therapy format (overall duration of EMDR treatment, duration of each treatment phase, type of SBA recommended, and therapeutic components addressing the various symptoms of disrupted self-organization [29]), (ii) the independent evaluation of the effectiveness of each of the different phases (preparatory phases and treatment phases), (iii) the characterization of the components included in group sessions (learning certain skills such as the “STOP” signal, peer support and experience sharing) compared with individual sessions and (iv) the evaluation of the impact of parenting interventions on reducing a child’s PTSD symptoms. Finally, as with prolonged exposure therapy, the evaluation of intensive [65] and brief [64] therapeutic programs incorporating EMDR would also be relevant for adolescents with complex PTSD due to its effectiveness, tolerability and high level of patient retention.

## Figures and Tables

**Figure 1 healthcare-12-01993-f001:**
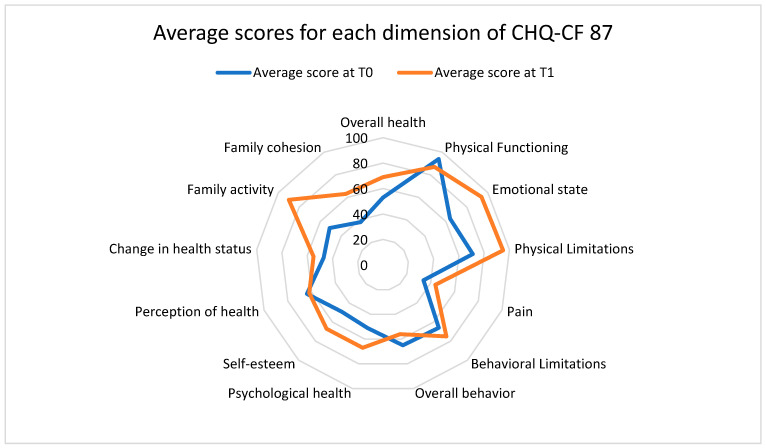
Average scores for each dimension of CHQ-CF 87. Legend: The figure shows the group’s average score (the TO average score is shown in blue; the T1 average score is shown in orange) on each of the 12 dimensions of the quality-of-life scale, based on a score ranging from 0 (worst possible quality of life) to 100 (best possible quality of life).

**Table 1 healthcare-12-01993-t001:** Socio-demographic characteristics of adolescent sample (N = 22).

Demographic Variables	Results
Sex, *n* (%)	
Female	16 (73%)
Male	6 (27%)
Age (years)	
Range	12–17
Mean	14.6
Type of trauma	
Physical violence	8 (36%)
Sexual violence	12 (54%)
Neglect	18 (82%)
Combination of different types of violence	19 (86%)
Parental perpetrators of abuse	19 (86%)
Age of onset of abuse	
Before age 5	18 (82%)
Between 5 and 10 years	4 (18%)
History of parental violence	15 (94%)
Family situation	
Lives with both parents	1 (4%)
Lives with one parent	18 (82%)
Foster family	3 (14%)
Child protection measures	
Ongoing child protection measures	8 (36%)
History of child protection actions	14 (64%)

Legend: Data are presented as number (percentage, %); age is presented in years.

**Table 2 healthcare-12-01993-t002:** EMDR treatment outcome measures.

Scale	N	Pre-Treatment T0		Post-Treatment T1		*p*
		Mean	SD	Mean	SD	
Child	21					
CPTS-RI		40.2	4.6	34.4	5.4	<0.001 ***
PDEQ		25.7	3.3	21.0	3.2	0.034
CDI		18.2	3.6	10.6	5.4	0.014 *
R-CMAS		21.3	5.5	13.3	6.3	0.004 **
ALS		29	11.3	10.8	9	0.024 *
ISI		18.5	4.4	9.2	5.4	0.001 **
CHQCF-87		57.9	9.8	77.4	13.6	0.005 **
ADOSPA		2.3	1	0.3	0.7	0.005 **
Parent	16					
CBCL 4–16		71.7	25.9	49.3	24.8	0.07
STAI		51.9	11	44	10.9	<0.001 ***

Legend: Data are presented as scores for each scale and quantitative variables are shown as mean (+/− standard deviation, SD). PTSD: post-traumatic stress disorder; CPTS-RI: Child Post-Traumatic Stress Reaction Index; Child Depression Inventory (CDI); Revised Children’s Manifest Anxiety Scale (R-CMAS); Affective Lability Scale (ALS); Insomnia Severity Index (ISI); Child Health Questionnaire CF87 (CHQ-CF87); Adolescents and Psychoactive Substances (ADOSPA) score; Child Behavior Checklist 4–16 (CBCL 4–16); State Trait Anxiety Inventory (STAI). * *p* < 0.05; ** *p* < 0.01; *** *p* < 0.001.

## Data Availability

Data are available upon request.

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
