# Peer review of "EMDR–Teens–cPTSD: Efficacy of Eye Movement Desensitization and Reprocessing in Adolescents with Complex PTSD Secondary to Childhood Abuse: A Case Series"

_healthcare, 2024, doi:10.3390/healthcare12191993_

Round 1
Reviewer 1 Report
Comments and Suggestions for Authors
Thank you for the opportunity to review an interesting article on the effectiveness of EMDR in adolescents suffering from cPTSD due to childhood abuse. I have only a few comments for the authors:
- The abstract is well written and informative.
- The introduction is sufficiently detailed and clearly structured. It might have been useful to expand the research objectives just a little.
- The materials and methods are clear and include all the necessary information. I am just wondering if the authors might have considered that there might be some differences between adolescents before and after COVID-19? Would it perhaps be useful to mention this in the limitations of the study?
- The results are well structured and insightful. Perhaps only the column widths in Table 1 could be corrected.
- The discussion is clear and informative and links the findings of the study well with the previous literature. It is important to me that the authors point out the absence of a control group as a limitation.
Author Response
Dear Reviewer 1,
Thank you for all your feedback and suggestions.
Review 1: The abstract is well written and informative.
Answer 1: Thank you for your appreciation.
Review 2: The introduction is sufficiently detailed and clearly structured. It might have been useful to expand the research objectives just a little.
Answer 2: Thank you for your suggestions to broaden the search objectives. We propose to add the evaluation of anxiety in parents as secondary objectives (see results in table 2: EMDR treatment outcomes measures). We also propose to formulate as tertiary objectives proposals for adjustments to the EMDR protocol detailed in section 4.3 of the discussion (see section 4.3 Therapeutic work with parents).
Initial paragraph: “The secondary objectives of the study were to evaluate the effectiveness of EMDR on the main comorbidities associated with adolescent PTSD symptoms such as depression, anxiety, emotional regulation disorders, sleep disorders, behavioral disorders and risky behavior, illicit substance consumption, but also its effectiveness on quality of life”.
Paragraph modified (line 167-169): “The secondary objectives of the study were to evaluate the effectiveness of EMDR on the main comorbidities associated with adolescent PTSD symptoms such as depression, anxiety, emotional regulation disorders, sleep disorders, behavioral disorders and risky behavior, illicit substance consumption, but also its effectiveness on quality of life and parental anxiety. The tertiary aims of the study were to develop proposals for adjustments to future EMDR protocols for adolescents suffering from post-traumatic stress disorder, with particular reference to work with parents”.
Review 3: The materials and methods are clear and include all the necessary information. I am just wondering if the authors might have considered that there might be some differences between adolescents before and after COVID-19? Would it perhaps be useful to mention this in the limitations of the study?
Answer 3: Yes, we have considered that there might be some differences between the groups treated before and after covid-19, due to the impact of covid-19 and confinement-related stress on adolescents' overall mental health. We therefore explored this factor during the assessment interviews and in the statistical analyses. There was no difference between the two groups. This astonishing phenomenon was explained by the young people by the fact that all the adolescents already benefited either from ongoing access to care with therapeutic support, or from a pre-existing socio-educational support network relating to their life course. We have included these elements in the text.
Sentence added to results section (line 361-362):“Finally, there was no difference in intra-group efficacy between adolescents treated before and after covid-19”.
Sentence added to discussion section (line 551-556): “Inclusion before and after covid-19 could have been a limitation. However, the lack of difference between the results of the included and treated groups before and after covid was explained by the young people by the fact that all the adolescents already benefited either from ongoing access to care with therapeutic support, or from a pre-existing socio-educational support network, and that covid was experienced as a minimal stressor in relation to previous life events”.
Review 4: The results are well structured and insightful. Perhaps only the column widths in Table 1 could be corrected.
Answer 4: The widths in table 1 have been modified for better visualization.
Review 5: The discussion is clear and informative and links the findings of the study well with the previous literature. It is important to me that the authors point out the absence of a control group as a limitation.
Answer 5: Thank you for your comments. indeed, this is the point we've given a great deal of thought to, weighing up the advantages and disadvantages according to the specific objectives of the study.
Initial paragraph: “The first limitation of the study is the absence of a control group benefiting from either another psychotherapy for cPTSD (such as the Integrative Treatment of Complex Trauma, ITCT (Briere & Lanktree, 2013) or a waiting list design”.
Discussion paragraph modified (line 508-529): “The absence of a control group in a study evaluating the efficacy of EMDR in adolescents with complex PTSD is the main limitation of our study. Indeed, without a no-therapy group, it is difficult to know whether the improvements observed are really due to EMDR's own effect or to the natural evolution of symptoms (spontaneous improvement bias), to external factors (family support or others) or to a placebo effect, which limits the generalizability of the results and their scientific acceptance. However, the aim of the study was not to demonstrate superiority between therapies, but rather to evaluate the efficacy of EMDR independently, and on criteria specific to complex PTSD. Intragroup follow-up (before/after EMDR) with standardized measurement tools and longitudinal evaluation may be sufficient to demonstrate individual progress, intervention efficacy, acceptability and tolerance without the need for a control group. Furthermore, from an ethical point of view, it is delicate to deprive severely affected adolescents of validated treatment or to randomize them to a therapy they have not chosen. Forcing participation in a group delivering a therapy other than that preferred by the patient could result in less adherence to treatment. Finally, although Tf-CBT is a validated approach for the treatment of PTSD, it is not necessarily considered superior to EMDR in all studies, and specifically concerning the treatment of PTSD-C (such as the Integrative Treatment of Complex Trauma, ITCT (Briere & Lanktree, 2013), no complex trauma therapy was available in the child psychiatric population in France at the time of the study. All in all, by focusing solely on EMDR, the study enables a direct assessment of its efficacy on a population and pa-thology of complex trauma with unique characteristics, while minimizing biases linked to ethical issues and non-adherence to treatment”.
Paragraph modified (line 564-566): “This study is preliminary study a larger randomized controlled trial, including a control group and blinded to group status”.
We would like to thank you once again for your feedback. We hope you find our corrections will be acceptable.
We are available if you feel that further modifications are necessary.

Reviewer 2 Report
Comments and Suggestions for Authors
The authors present the results of a trial assessing the efficacy of eye movement densensitisation and reprocessing (EDMR) on adolescence complex posttraumatic stress disorder. They conclude that "EMDR is a highly promising treatment method for complex PTSD".
The research design is seriously flawed. This is a relatively small (n = 22) study which lacks a control group and which is therefore unblinded. I strongly recommend that the authors consider carrying out a larger, randomised controlled trial of EMDR in this condition; the assessments and analyses should be conducted blind to group status.
Author Response
Dear Reviewer 2,
Thank you for all your feedback and suggestions.
Review: The authors present the results of a trial assessing the efficacy of eye movement densensitisation and reprocessing (EDMR) on adolescence complex posttraumatic stress disorder. They conclude that "EMDR is a highly promising treatment method for complex PTSD".
The research design is seriously flawed. This is a relatively small (n = 22) study which lacks a control group and which is therefore unblinded. I strongly recommend that the authors consider carrying out a larger, randomised controlled trial of EMDR in this condition; the assessments and analyses should be conducted blind to group status.
Answer: Thank you for your analysis of the article and your comments. Indeed, the lack of a control group and blind analysis is the study's weak point. This methological aspect given a great deal of thought to, weighing up the advantages and disadvantages according to the specific objectives of the study. As a result, we have modified the limits section of the discussion and the conclusion in the abstract and in the text, moderating the results obtained and emphasizing the research recommendations. We have recommended that consideration be given to a larger randomized controlled trial evaluating the efficacy of EMDR in adolescents with cPTSD versus a control group.
Initial abstract conclusion (line 31-36): “Our study shows that EMDR is a highly promising treatment method for complex PTSD, reducing posttraumatic symptoms and certain comorbid disorders frequently in adolescents with a history of abuse in childhood. More research is needed on adolescent populations with cPTSD (e.g., controlled EMDR trials, comparison to other therapies, evaluation of the action of the different phases of the protocol)”.
Abstract conclusion modified (line 31-36): “The results of this study are encouraging and suggest that EMDR may be effective in the symptom management reducing post-traumatic symptoms and certain comorbid disorders frequently seen in adolescents who have experienced childhood abuse. Further research is needed on adolescent populations suffering from cPTSD (e.g, randomized controlled trials with control groups and other therapies, or evaluating the action of the different phases of the protocol)”.
Initial discussion paragraph (line 499-502): “The first limitation of the study is the absence of a control group benefiting from either another psychotherapy for cPTSD (such as the Integrative Treatment of Complex Trauma, ITCT (Briere & Lanktree, 2013) or a waiting list design”.
Discussion paragraph modified (line 508-529): “The absence of a control group in a study evaluating the efficacy of EMDR in adolescents with complex PTSD is the main limitation of our study. Indeed, without a no-therapy group, it is difficult to know whether the improvements observed are really due to EMDR's own effect or to the natural evolution of symptoms (spontaneous improvement bias), to external factors (family support or others) or to a placebo effect, which limits the generalizability of the results and their scientific acceptance. However, the aim of the study was not to demonstrate superiority between therapies, but rather to evaluate the efficacy of EMDR independently, and on criteria specific to complex PTSD. Intragroup follow-up (before/after EMDR) with standardized measurement tools and longitudinal evaluation may be sufficient to demonstrate individual progress, intervention efficacy, acceptability and tolerance without the need for a control group. Furthermore, from an ethical point of view, it is delicate to deprive severely affected adolescents of validated treatment or to randomize them to a therapy they have not chosen. Forcing participation in a group delivering a therapy other than that preferred by the patient could result in less adherence to treatment. Finally, although Tf-CBT is a validated approach for the treatment of PTSD, it is not necessarily considered superior to EMDR in all studies, and specifically concerning the treatment of PTSD-C (such as the Integrative Treatment of Complex Trauma, ITCT (Briere & Lanktree, 2013), no complex trauma therapy was available in the child psychiatric population in France at the time of the study. All in all, by focusing solely on EMDR, the study enables a direct assessment of its efficacy on a population and pa-thology of complex trauma with unique characteristics, while minimizing biases linked to ethical issues and non-adherence to treatment”.
Discussion paragraph added (line 564-566): “This study is preliminary study a larger randomized controlled trial, including a control group and blinded to group status”.
Initial conclusion paragraph (line 561-580): “The treatment of these adolescents made it possible to suggest adaptations of the EMDR protocol to complex trauma which should be evaluated, in particular by comparing the effectiveness of EMDR to other therapies targeted at complex PTSD, and the integration of EMDR to other multimodal therapies.
Future research on EMDR in complex trauma should focus on standardizing (i) the format of the therapy (overall duration of EMDR treatment, duration of each phase of treatment, type of recommended SBA, therapeutic components treating symptoms dis-ruption of the organization of the self [29], (ii) the independent evaluation of the effec-tiveness of each of the different phases (preparatory phases and treatment phases), (iii) the characterization of the components included in group sessions (‘STOP’ signal learning in group, peer experience sharing and support, normalization of group experience, etc.) versus individual sessions and (iv) the evaluation of the impact of parental interventions on the reduction of the child's cPTSD symptoms. These study results need to be replicated in larger equivalent populations by evaluating the effects of EMDR over the long term, and with comparison groups treated with alternative therapies targeted at cPTSD such as Integrative Treatment of Complex Trauma for Adolescents (ITCT-A) [73]. Finally, as for prolonged exposure therapy, the evaluation of intensive [65] and brief [64] therapeutic program integrating EMDR would also be relevant for adolescents with complex PTSD due to its effectiveness, tolerability and high level of patient retention”.
Conclusion paragraph modified (line 579-593): “The results of these studies should be replicated on larger equivalent populations by evaluating the effects of EMDR to control groups (without or with therapies targeting cPTSD such as Integrative Treatment of Complex Trauma for Adolescents (ITCT-A)), and in the long term [73]. More generally, future research on EMDR in complex trauma should incorporate protocol adaptations focusing on standardizing (i) the therapy format (overall duration of EMDR treatment, duration of each treatment phase, type of SBA recommended, therapeutic components addressing the various symptoms of disrupted self-organization [29]), (ii) independent evaluation of the effectiveness of each of the different phases (preparatory phases and treatment phases), (iii) characterization of the components included in group sessions (learning certain skills such as the “STOP” signal, peer support and experience sharing) compared with individual sessions and (iv) evaluation of the impact of parenting interventions on reducing the child's PTSD symptoms. Finally, as with prolonged exposure therapy, the evaluation of intensive [65] and brief [64] therapeutic programs incorporating EMDR would also be relevant for adolescents with complex PTSD due to its effectiveness, tolerability and high level of patient retention”.
We would like to thank you once again for your feedback. We hope you find our corrections will be acceptable.
We are available if you feel that further modifications are necessary.

Reviewer 3 Report
Comments and Suggestions for Authors
Thanks to the authors for sharing their manuscript. I have two serious concerns about the design of the study:
· The study participants were adolescents who received interventions. Why was there no control group with participants who did not receive intervention? The absence of a control group casts doubt on the reliability of the conclusions about the effectiveness of the intervention.
· The authors took two measurements, before the intervention and six weeks after the interventions. The question arises: why did the authors not take measurements immediately after the last intervention session? This point should be explained in detail, because without measurement immediately after the interventions, it becomes unclear whether the interventions accurately affected the outcomes.
Other comments:
· In the Method, a description of all the measures is given in one paragraph. It is difficult to perceive information in this form, I recommend that the authors divide the description of the measures into different paragraphs and highlight their names in bold.
· In the abstract, the authors write that the repeated measurement was carried out six weeks after the first measurement (“Subjective measures of PTSD and associated psychiatric disorders were taken before (T0) and after 6 weeks of EMDR therapy (T1)”; lines 23-24), and then they write that three months later (“The initial assessment was carried out one week before therapy (T0) and the post-treatment assessment (T1) 3 months after the end of total EMDR therapy (6th EMDR session)”; lines 214-216).
I wish the authors good luck in revising the manuscript and hope that my comments will help them in this.
Sincerely yours,
the Reviewer.
Author Response
Dear Reviewer 3,
Thank you for all your feedback and suggestions.
Review 1: The study participants were adolescents who received interventions. Why was there no control group with participants who did not receive intervention? The absence of a control group casts doubts on the reliability of the conclusions about the effectiveness of the intervention.
Answer 1: Thank you for your analysis of the article and your comments. Indeed, the lack of a control group and blind analysis is the study's weak point. This methological aspect given a great deal of thought to, weighing up the advantages and disadvantages according to the specific objectives of the study. As a result, we have modified the limits section of the discussion and the conclusion in the abstract and in the text, moderating the results obtained and emphasizing the research recommendations. We have recommended that consideration be given to a larger randomized controlled trial evaluating the efficacy of EMDR in adolescents with cPTSD versus a control group.
Initial abstract conclusion (line 31-36): “Our study shows that EMDR is a highly promising treatment method for complex PTSD, reducing posttraumatic symptoms and certain comorbid disorders frequently in adolescents with a history of abuse in childhood. More research is needed on adolescent populations with cPTSD (e.g., controlled EMDR trials, comparison to other therapies, evaluation of the action of the different phases of the protocol)”.
Abstract conclusion modified (line 31-36): “The results of this study are encouraging and suggest that EMDR may be effective in the symptom management reducing post-traumatic symptoms and certain comorbid disorders frequently seen in adolescents who have experienced childhood abuse. Further research is needed on adolescent populations suffering from cPTSD (e.g, randomized controlled trials with control groups and other therapies, or evaluating the action of the different phases of the protocol)”.
Initial discussion paragraph (line 499-502): “The first limitation of the study is the absence of a control group benefiting from either another psychotherapy for cPTSD (such as the Integrative Treatment of Complex Trauma, ITCT (Briere & Lanktree, 2013) or a waiting list design”.
Discussion paragraph modified (line 508-529): “The absence of a control group in a study evaluating the efficacy of EMDR in adolescents with complex PTSD is the main limitation of our study. Indeed, without a no-therapy group, it is difficult to know whether the improvements observed are really due to EMDR's own effect or to the natural evolution of symptoms (spontaneous improvement bias), to external factors (family support or others) or to a placebo effect, which limits the generalizability of the results and their scientific acceptance. However, the aim of the study was not to demonstrate superiority between therapies, but rather to evaluate the efficacy of EMDR independently, and on criteria specific to complex PTSD. Intragroup follow-up (before/after EMDR) with standardized measurement tools and longitudinal evaluation may be sufficient to demonstrate individual progress, intervention efficacy, acceptability and tolerance without the need for a control group. Furthermore, from an ethical point of view, it is delicate to deprive severely affected adolescents of validated treatment or to randomize them to a therapy they have not chosen. Forcing participation in a group delivering a therapy other than that preferred by the patient could result in less adherence to treatment. Finally, although Tf-CBT is a validated approach for the treatment of PTSD, it is not necessarily considered superior to EMDR in all studies, and specifically concerning the treatment of PTSD-C (such as the Integrative Treatment of Complex Trauma, ITCT (Briere & Lanktree, 2013), no complex trauma therapy was available in the child psychiatric population in France at the time of the study. All in all, by focusing solely on EMDR, the study enables a direct assessment of its efficacy on a population and pathology of complex trauma with unique characteristics, while minimizing biases linked to ethical issues and non-adherence to treatment”.
Discussion paragraph added (line 564-566): “This study is preliminary study a larger randomized controlled trial, including a control group and blinded to group status”.
Initial conclusion paragraph (line 561-580): “The treatment of these adolescents made it possible to suggest adaptations of the EMDR protocol to complex trauma which should be evaluated, in particular by comparing the effectiveness of EMDR to other therapies targeted at complex PTSD, and the integration of EMDR to other multimodal therapies.
Future research on EMDR in complex trauma should focus on standardizing (i) the format of the therapy (overall duration of EMDR treatment, duration of each phase of treatment, type of recommended SBA, therapeutic components treating symptoms disruption of the organization of the self [29], (ii) the independent evaluation of the effectiveness of each of the different phases (preparatory phases and treatment phases), (iii) the characterization of the components included in group sessions (‘STOP’ signal learning in group, peer experience sharing and support, normalization of group experience, etc.) versus individual sessions and (iv) the evaluation of the impact of parental interventions on the reduction of the child's cPTSD symptoms. These study results need to be replicated in larger equivalent populations by evaluating the effects of EMDR over the long term, and with comparison groups treated with alternative therapies targeted at cPTSD such as Integrative Treatment of Complex Trauma for Adolescents (ITCT-A) [73]. Finally, as for prolonged exposure therapy, the evaluation of intensive [65] and brief [64] therapeutic program integrating EMDR would also be relevant for adolescents with complex PTSD due to its effectiveness, tolerability and high level of patient retention”.
Conclusion paragraph modified (line 579-593): “The results of these studies should be replicated on larger equivalent populations by evaluating the effects of EMDR to control groups (without or with therapies targeting cPTSD such as Integrative Treatment of Complex Trauma for Adolescents (ITCT-A)), and in the long term [73]. More generally, future research on EMDR in complex trauma should incorporate protocol adaptations focusing on standardizing (i) the therapy format (overall duration of EMDR treatment, duration of each treatment phase, type of SBA recommended, therapeutic components addressing the various symptoms of disrupted self-organization [29]), (ii) independent evaluation of the effectiveness of each of the different phases (preparatory phases and treatment phases), (iii) characterization of the components included in group sessions (learning certain skills such as the “STOP” signal, peer support and experience sharing) compared with individual sessions and (iv) evaluation of the impact of parenting interventions on reducing the child's PTSD symptoms. Finally, as with prolonged exposure therapy, the evaluation of intensive [65] and brief [64] therapeutic programs incorporating EMDR would also be relevant for adolescents with complex PTSD due to its effectiveness, tolerability and high level of patient retention”.
Review 2: The authors took two measurements, before the intervention and six weeks after the interventions. The question arises: why did the authors not take measurements immediately after the last intervention session? This point should be explained in detail, because without measurement immediately after the interventions, it becomes unclear whether the interventions accurately affected the outcomes.
Answer 2: Evaluating the effectiveness of EMDR six weeks after the sessions enables us to measure the durability of the therapeutic effects. The benefits of EMDR, notably the reprocessing of traumatic memories, can continue to be reinforced over time, beyond the end of the sessions. By avoiding the immediate effect of therapy, this delayed evaluation better captures therapeutic consolidation and the integration of learning into patients' daily lives. It also checks whether results persist without the lingering effect of proximity to the therapist, ensuring that improvements are stable and autonomous, rather than influenced by immediate therapeutic attention.
Discussion paragraph added (line 329-339): The second limitation is the absence of long-term follow-up measures which does not allow us to conclude on the maintenance of the beneficial effect of EMDR over time [74]. However, to compensate for this effect, we have chosen to evaluate the effectiveness of EMDR six weeks after the sessions, in order to measure the durability of the therapeutic effects. The benefits of EMDR, notably the reprocessing of traumatic memories, can continue to be reinforced over time, beyond the end of the sessions. By avoiding the immediate effect of therapy, this delayed evaluation better captures therapeutic consolidation and the integration of learning into patients' daily lives. It also checks whether results persist without the lingering effect of proximity to the therapist, ensuring that improvements are stable and autonomous, rather than influenced by immediate therapeutic attention.
Review 3: In the Method, a description of all the measures is given in one paragraph. It is difficult to perceive information in this form, I recommend that the authors divide the description of the measures into different paragraphs and highlight their names in bold.
Answer 3: Thank you for your suggestion to divide the paragraphs into different sections, which makes the evaluations easier to read.
Initial paragraph (line 201-246): “2.2. Measures
All sociodemographic data of adolescent subjects (age, sex, place of accommodation) were extracted from medical-administrative document completed upon admission to the child and adolescent psychiatry department. Medical clinical interviews assessed traumatic exposures such as nature of traumatic events, date and frequency of traumatic exposure, and parental history of exposure to traumatic events. Information on family situation and protection measures (history of protection measures or current protection measures) were also collected. The effectiveness of EMDR on posttraumatic symptoms and comorbid disorders was evaluated using validated self-questionnaires completed by the adolescent and by one parent when available. The initial assessment was carried out one week before therapy (T0) and the post-treatment assessment (T1) 3 months after the end of total EMDR therapy (6th EMDR session). The impact of EMDR on posttraumatic symptoms was assessed at T0 and T1 using (i) the Child Posttraumatic Stress Reaction Index (CPTS-RI) and the Peritraumatic Dissociative Experiences Questionnaire (PDEQ). The Child Post-Traumatic Stress Reaction Index (CPTS-RI) is a 20- item self-questionnaire used to confirm the diagnosis of PTSD, and its level of severity. A total score of 12-24 is associated with mild PTSD, a score of 25-39 with moderate level, a score of 40-59 with severe level, and a score above 60 with very severe level. The Cronbach alpha coefficient is 0.87 for the French version of the CPTS-RI [33]. The Peritraumatic Dissociative Experiences Questionnaire (PDEQ) [34] measures the experiences of dissociation experienced with 10 items assessing the degree of depersonalization, unreality, amnesia, modification of time perception and modification of one's body image. A score greater than 15 indicates significant peritraumatic dissociation. The impact of EMDR (difference at T0 and T1) on PTSD comorbidities was assessed using (ii) the Child Depression Inventory (CDI) to quantify depressive symptoms (score ranges from 0 to 54: the higher the score, the more severe the level of depression [35], (iii) the Re-vised-Children's Manifest Anxiety Scale (R-CMAS) to quantify anxiety symptoms [36], (iv) the Affective Lability Scale (ALS) to quantify emotional regulation difficulties [37], (v) the Insomnia Severity Index (ISI) to assess insomnia (total score specifies the degree of insomnia: absence of insomnia (0-7), symptoms of subclinical insomnia (8-14), moderate insomnia (15-21), and severe insomnia (22-28)) [38], (vi) the Child Health Questionnaire CF87 (CHQ-CF87) to assess at different dimensions (physical functioning, emotional state, physical limitations, pain, behavioral limitations, overall behavior, mental health, self-esteem, perception of health, change in state health, family activity and family cohesion) of quality of life (dimensions scored on a scale ranging from 0 (worst possible quality of life) to 100 (best possible quality of life) [39], (vii) the Adolescents and Psychoactive Substances (ADOSPA) to quantify the harmful use of alcohol and drugs (low risk (0-1 point), risk moderate (2 points) and high risk (≥ 3)) [40], and (viii) the Child Behavior Checklist 4-16 (CBCL 4-16) to evaluate the child's behavior based on parental observation (sensitivity threshold score at 40 for boys and 37 for girls) [41]. Parental anxiety was assessed with (xi) the administration of the State-Trait Anxiety Inventory (STAI) [42] in order to obtain indirect information on the parents' psychological state and their possible possibilities for emotional regulation. The total score indicates the severity of the general state of anxiety with a rating of 20 to 80. A score between 36 to 45 indicates low anxiety, a score between 46 and 55 indicates moderate anxiety, a score between 56 to 65 indicates high anxiety and a score above 65 indicates very high anxiety”.
Paragraph modified (line 201-256): “2.2. Measures
Socio-demographic and clinical characteristics of adolescent and his family
All sociodemographic data of adolescent subjects (age, sex, place of accommodation) were extracted from medical-administrative document completed upon admission to the child and adolescent psychiatry department. Medical clinical interviews assessed traumatic exposures such as nature of traumatic events, date and frequency of traumatic exposure, and parental history of exposure to traumatic events. Information on family situation and protection measures (history of protection measures or current protection measures) were also collected.
Assessment of adolescent PTSD symptoms
The effectiveness of EMDR on posttraumatic symptoms and comorbid disorders was evaluated using validated self-questionnaires completed by the adolescent and by one parent when available. The initial assessment was carried out one week before therapy (T0) and the post-treatment assessment (T1) 3 months after the end of total EMDR therapy (6th EMDR session).
The impact of EMDR on posttraumatic symptoms was assessed at T0 and T1 using (i) the Child Posttraumatic Stress Reaction Index (CPTS-RI) and the Peritraumatic Dissociative Experiences Questionnaire (PDEQ). The Child Post-Traumatic Stress Reaction Index (CPTS-RI) is a 20- item self-questionnaire used to confirm the diagnosis of PTSD, and its level of severity. A total score of 12-24 is associated with mild PTSD, a score of 25-39 with moderate level, a score of 40-59 with severe level, and a score above 60 with very severe level. The Cronbach alpha coefficient is 0.87 for the French version of the CPTS-RI [33]. The Peritraumatic Dissociative Experiences Questionnaire (PDEQ) [34] measures the experiences of dissociation experienced with 10 items assessing the degree of depersonalization, unreality, amnesia, modification of time perception and modification of one's body image. A score greater than 15 indicates significant peritraumatic dissociation.
Assessment of PTSD comorbidities in adolescents and parents
The impact of EMDR (difference at T0 and T1) on PTSD comorbidities was assessed using (ii) the Child Depression Inventory (CDI) to quantify depressive symptoms (score ranges from 0 to 54: the higher the score, the more severe the level of depression [35], (iii) the Revised-Children's Manifest Anxiety Scale (RCMAS) to quantify anxiety symptoms [36], (iv) the Affective Lability Scale (ALS) to quantify emotional regulation difficulties [37], (v) the Insomnia Severity Index (ISI) to assess insomnia (total score specifies the degree of insomnia: absence of insomnia (0-7), symptoms of subclinical insomnia (8-14), moderate insomnia (15-21), and severe insomnia (22-28)) [38], (vi) the Child Health Questionnaire CF87 (CHQ-CF87) to assess at different dimensions (physical functioning, emotional state, physical limitations, pain, behavioral limitations, overall behavior, mental health, self-esteem, perception of health, change in state health, family activity and family cohesion) of quality of life (dimensions scored on a scale ranging from 0 (worst possible quality of life) to 100 (best possible quality of life) [39], (vii) the Adolescents and Psychoactive Substances (ADOSPA) to quantify the harmful use of alcohol and drugs (low risk (0-1 point), risk moderate (2 points) and high risk (≥ 3)) [40], and (viii) the Child Behavior Checklist 4-16 (CBCL 4-16) to evaluate the child's behavior based on parental observation (sensitivity threshold score at 40 for boys and 37 for girls) [41]. Parental anxiety was assessed with (xi) the administration of the State-Trait Anxiety Inventory (STAI) [42] in order to obtain indirect information on the parents' psychological state and their possible possibilities for emotional regulation. The total score indicates the severity of the general state of anxiety with a rating of 20 to 80. A score between 36 to 45 indicates low anxiety, a score between 46 and 55 indicates moderate anxiety, a score between 56 to 65 indicates high anxiety and a score above 65 indicates very high anxiety”.
Review 4: In the abstract, the authors write that the repeated measurement was carried out six weeks after the first measurement (“Subjective measures of PTSD and associated psychiatric disorders were taken before (T0) and after 6 weeks of EMDR therapy (T1)”; lines 23-24), and then they write that three months later (“The initial assessment was carried out one week before therapy (T0) and the post-treatment assessment (T1) 3 months after the end of total EMDR therapy (6th EMDR session)”; lines 214-216).
Answer 4: Thank you for your attention. It's a mistake: it's 3 months. >e've made the appropriate corrections in the summary and text.
We would like to thank you once again for your feedback. We hope you find our corrections will be acceptable.
We are available if you feel that further modifications are necessary.

Round 2
Reviewer 2 Report
Comments and Suggestions for Authors
I thank the authors for their response to my previous comments.
Reviewer 3 Report
Comments and Suggestions for Authors
Thank you for sharing the revised manuscript. I confirm that all my comments have been taken into account, and I believe that the manuscript can be recommended for publication.